# Increasing the Photocatalytic Activity of BiVO$_4$ by Naked Co(OH)$_2$ Nanoparticle Cocatalysts

Luiz E. Gomes [1], Luiz F. Plaça [1], Washington S. Rosa [2], Renato V. Gonçalves [2], Sajjad Ullah [1,3] and Heberton Wender [1,*]

1    Laboratory of Nanomaterials and Applied Nanotechnology, Nano & Photon Research Group, Institute of Physics, Federal University of Mato Grosso do Sul, Campo Grande 79070-900, Brazil
2    Instituto de Física de São Carlos, Universidade de São Paulo, P.O. Box 369, São Carlos 13560-970, Brazil
3    Institute of Chemical Sciences, University of Peshawar, Peshawar P.O. Box 25120, Pakistan
*    Correspondence: heberton.wender@ufms.br

**Abstract:** Bismuth vanadate (BiVO$_4$ or BVO) is one of the most studied photocatalysts for water oxidation because of its excellent visible light absorption and appropriate band energy positions. However, BVO presents a low charge mobility and a high electron–hole recombination rate. To address these fundamental limitations, this study proposes the coating of previously synthesized phase-pure monoclinic scheelite BVO with different amounts of naked cobalt (further oxidized to cobalt hydroxide) nanoparticles (NPs) via a modified magnetron sputtering deposition. The resulting BVO/Co photocatalysts were investigated for methylene blue (MB) photodegradation, photocatalytic oxygen evolution, and photoelectrochemical (PEC) water oxidation. In the MB photodegradation tests, the BVO/Co sample prepared with a deposition time of 5 min (BVO/Co(5 min)) presented the highest photoactivity (k = 0.06 min$^{-1}$) compared with the other sputtering investigated times (k = 0.01–0.02 min$^{-1}$), as well as the pristine BVO sample (k = 0.04 min$^{-1}$). A similar trend was evidenced for the PEC water oxidation, where a photocurrent density of 23 µA.cm$^{-2}$ at 1.23 V (vs. RHE) was observed for the BVO/Co(5 min) sample, a value 4.6 times higher compared with pristine BVO. Finally, the BVO/Co(5 min) presented an O$_2$ evolution more than two times higher than that of the pristine BVO. The increased photocatalytic performance was ascribed to increased visible-light absorption, lesser electron–hole recombination, and enhanced charge transfer at the liquid/solid interface. The deposition of Co(OH)$_2$ NPs via magnetron sputtering can be considered an effective strategy to improve the photocatalytic performance of BVO for different target catalytic reactions, including oxygen evolution, water oxidation, and pollutant photodegradation.

**Keywords:** oxygen evolution; photoelectrochemical water splitting; cocatalyst; cobalt hydroxide; sputtering; photocatalysis





## 1. Introduction

The accelerated growth in the world's population and the accompanying increase in industrialization has not only stressed the global hydrocarbon-based energy reservoirs but has also caused the serious problem of environmental pollution. From a sustainability point of view, it is urgent to find effective strategies to address these global challenges, for example, by searching for alternative energy sources and designing effective environmental remediation processes [1,2]. Thus, a great deal of scientific interest has been directed towards the exploitation of the abundant and never-exhausting solar light for photo(electro)chemical processes aimed at generating/storing greener energy and/or combating environmental pollution [3,4].

Heterogeneous photocatalysis, which employs semiconductor-based materials for sunlight capture and the consequent formation of charge carriers (electrons and holes with a high redox potential) is considered one of the most promising strategies to effectively utilize

solar energy for different photo(electro)chemical applications [5–9]. For instance, photoelectrochemical (PEC) water splitting using semiconductor photoelectrodes can directly convert solar energy into fuels in a green and sustainable way [10,11]. A suitable photocatalyst for efficient solar-to-electrical/chemical energy conversion must exhibit a good absorption of solar light, as well as efficient generation, transport, and utilization of photogenerate charge carriers [1,10,12]. Unfortunately, no single photocatalytic material fulfills all of the thermodynamic requirements and most of the relatively efficient photocatalysts ($TiO_2$ and ZnO) only operate under UV light, which only accounts for a limited region of the solar spectrum [2,11]. Thus, many studies have focused on the use of mixed/composite materials that combine the advantages of the individual components into one system, often in a synergic manner [13–19], and the development of efficient visible/near-infrared light absorber material to improve the overall solar-to-electro(chemical) energy conversion [4,11,20,21].

Among the visible light photocatalysts, $BiVO_4$ (BVO), an n-type semiconductor with a band gap (~2.5 eV), high visible light absorption, and appropriate band structure/edge position, has attracted special attention for $CO_2$ reduction [2], water-splitting [11,14], and the photo-oxidation of organic molecules [4,21,22]. Unfortunately, however, it suffers from fast electron–hole recombination, poor electron mobility, and slow surface reaction kinetics for oxygen evolution, factors that ultimately limit the solar-to-energy conversion efficiency of BVO photoanodes/materials. To address these fundamental limitations of BVO and enhance its photocatalytic activity, different strategies have been proposed, including the formation of heterojunctions [4,23–25], metal-doping/decoration [22,26], and the deposition of cocatalyst materials on its surface [13–15,27–29]. The loading of cobalt-based oxygen evolution cocatalysts has proven to be a valuable method to enhance BVO oxidation ability as it accelerates the charge carrier transfer at the BVO surface. It is known that different structural forms of cobalt such as oxides, hydrated oxides, hydroxides, and oxyhydroxides might evolve in oxidation reaction conditions, which makes this class of materials of special interest as cocatalysts [30]. After the heterostructure formation of BVO with $Co(OH)_2$ through a chemical impregnation method, the photocurrent density for water splitting was significantly enhanced from 1.57 to 4.52 mA/cm$^2$ at 1.23 V vs. RHE under 1-sun illumination [31]. In parallel, a photocurrent density improvement for water oxidation was also obtained after depositing a thin $Co(OH)_2$ cocatalyst layer over BVO through atomic layer deposition [30]. More than that, the authors showed that the $Co^{2+}$ hydroxide performed better as a cocatalyst for BVO than the $Co^{2+}$ oxide. In another study, loading of BVO with the $Co(OH)_2$ cocatalyst lead to increased photoactivity for Rhodamine B degradation [32]. Although the $BiVO_4$/$Co(OH)_2$ photocatalytic system has already been reported, the enhancement in photoactivity is not well understood, as revealed by the different and controversial mechanisms proposed in the literature [30–32].

In this scenario, the present study proposes, for the first time, a solution-free controlled deposition method to coat pre-synthesized monoclinic scheelite BVO particles with different amounts of cobalt hydroxide nanoparticles (NPs) via magnetron sputtering using different deposition times in a special vibrating apparatus. The novel idea of the use of the vibrating apparatus ensures a homogenous coating of BVO particles with plentiful active sites, and the deposition method does not require the use of toxic solvents. Moreover, the amount/loading of NPs can be easily controlled by simply varying the deposition time. This allowed for a systematic study of the loading/layer thickness of the deposited NPs on the photoactivity of BVO/Co materials, as measured by methylene blue (MB) photodegradation, photocatalytic oxygen evolution, and photoelectrochemical (PEC) water oxidation. We thus provide important insights into the structure–property relationship of the BVO/Co materials, demonstrating that the deposition of $Co(OH)_2$ NPs increase visible-light absorption, decrease charge transfer resistance at the liquid/solid interface, and mitigate electron/hole recombination, boosting the photocatalytic performance of BVO in the above-intoned applications. The developed magnetron sputtering-based strategy for the deposition of Co-based NPs is general and substrate-independent and can be applied to homogenously modify the surface of different photocatalytic materials with a variety of

desired cocatalysts particles, facilitating preparation of photocatalyst/cocatalyst systems with improved photocatalytic performances.

## 2. Materials and Methods

### 2.1. Chemicals

Bismuth(III) nitrate pentahydrate ($Bi(NO_3)_3.5H_2O$, 98%), ferric nitrate nonahydrate ($Fe(NO_3)_3.9H_2O$, >98%), and ammonium metavanadate ($NH_4VO_3$, 99%) were purchased from Sigma Aldrich Chemical Co., Brazil. Methylene blue (MB) was obtained from Vetec, Campo Grande, Brazil. All of the solvents were of analytical grade and were used without further purification.

### 2.2. Synthesis of BVO Nanoflakes

The synthesis of the BVO powders was conducted as previously reported by our group, with no modifications [22,33]. Briefly, 8 mmol of $Bi(NO_3)_3.5H_2O$ and 8 mmol of $NH_4VO_3$ were separately dissolved into 11 mL of nitric acid (2.0M, called SA) and 19.8 mL of sodium hydroxide (2.0 M, called SB), respectively. These solutions were kept under sonication for 30 min (UltraCleaner® 1400A 40 kHz) and, in the sequence, magnetically stirred for another 30 min. After complete solubilization, SA was dropwise added into SB under stirring and a yellow solution was obtained. The final solution was magnetically stirred for 150 min and the pH was adjusted to 5.5 during the process. The reaction mixture was transferred into a 110 mL Teflon-lined stainless-steel autoclave and carefully sealed. The hydrothermal reaction was performed inside an oven at 140 °C for 6 h with a heating rate of 5 °C min$^{-1}$. After cooling, the final precipitate was centrifuged at 4500 rpm for 15 min, and washed four times with deionized water and once with ethyl alcohol. The obtained powder was dried at 60 °C for 12 h in air and stored in an Eppendorf.

### 2.3. Cobalt Cocatalyst Deposition

Modified magnetron sputtering deposition was employed to deposit the CoNPs on the surface of the BVO. Typically, 50 mg of the pristine BVO powder was placed in a round-bottom glass capsule connected to a mechanical resonant agitator, as previously reported [34,35]. This whole set was inserted in the sputtering chamber (5 cm from the target), which was evacuated to a base pressure of $2.0 \times 10^{-6}$ mbar. A working pressure of $3.5 \times 10^{-2}$ mbar was set up with the addition of ultra-pure argon (99.999%) into the chamber. The powder was agitated inside the chamber with the aid of a sinusoidal wave generation supply vibrating at a frequency of 90–100 Hz. A 100 W DC was applied at a cobalt metallic target (99.95%) at different sputtering times (5, 15, and 30 min) to produce samples with varying amounts of Co. The resulting samples were coded as BVO/Co($x$ min) where $x$ = 5, 15, and 30 min, representing the Co deposition time.

### 2.4. Photoelectrodes Preparation

Thin films were prepared by drop-casting 220 µL of a deionized water suspension of the samples (5 mg.mL$^{-1}$) onto a $1 \times 1$ cm selected area of a pre-cleaned glass substrate coated with Sn-doped indium oxide (ITO). After naturally drying, the films were annealed in Ar for 2 h at 300 °C at a rate of 5 °C.min$^{-1}$. Before sample deposition, all of the substrates were cleaned in an ultrasonic bath with deionized water and detergent, followed by immersion in ethyl alcohol and acetone for 20 min each, and were then dried at ambient temperature.

### 2.5. Characterizations

The pristine sample morphology and size were studied using a Sigma scanning electron microscope (Zeiss, Oberkochen, Germany) equipped with a field emission gun (FEG-SEM) and a high-resolution transmission electron microscope (JEOL-JEM 2100F), equipped with a field emission gun and operated at 200 kV. X-ray powder diffraction (XRD) patterns were recorded using a Bruker X-ray diffractometer with Cu Kα radiation,

$20 \leq 2\theta \leq 80$ range, angular step of $0.02°$, and a counting time of 5 s per step. The chemical surface composition of the samples was studied by X-ray photoelectron spectroscopy (XPS) using a conventional XPS spectrometer (ESCA+, Scientia Omicron, Uppsala, Sweden) with a high-performance hemispheric analyzer (EAC2000) and monochromatic Al K$\alpha$ radiation ($h\nu$ = 1486.6 eV) as the excitation source. The operating pressure in the ultra-high vacuum chamber (UHV) during the analysis was $10^{-9}$ Pa. The XPS high-resolution spectra were recorded at a constant passing energy of 20 eV with a 0.05 eV per step. The XPS spectra were processed using the Casa XPS software (release 2.3.16, Casa Software Ltd., London, UK). A Shirley background subtraction was applied and the peak positions of C1s carbon adventitious were used for energy calibration at 284.8 eV. Valence band XPS data were measured in a high-resolution mode and calibrated by linear extrapolation of the signal to zero intensity. UV–VIS diffuse reflectance spectra of the powders were obtained using a LAMBDA 650 UV–VIS spectrometer (PerkinElmer, Waltham, MA, USA) equipped with an integrating sphere and were converted from reflection to absorption by the Kubelka–Munk method. The Raman spectra were recorded by a Raman spectrometer (SCIAPS, Advantage 532, Woburn, MA, USA), using a 532-nm laser source for excitation. Photoluminescence (PL) spectra were recorded using a Fluorolog-3.11 spectrofluorometer (Horiba Jobin Yvon, Paris, France) equipped with a 450 W ozone-free Xenon lamp and a photomultiplier detector sensitive in the range of 200–850 nm, under excitation at 375 nm.

### 2.6. PEC Water Oxidation Measurements

PEC measurements were recorded using a CorrTest® potentiostat (Wuhan, China), with a standard three-electrode system. The as-prepared sample-based electrodes were used as the working electrode, an Ag/AgCl (saturated KCl) as the reference electrode, and a platinum rod as the counter electrode, all immersed in a 0.5 M $Na_2SO_4$ electrolyte saturated with Ar for 15 min to remove the dissolved oxygen. Linear scan voltammetry (LSV) was conducted in the dark and under irradiation from a solar simulator equipped with a 150 W xenon lamp (Model 10500, Abet Tech, Milford, CT, USA) and an AM 1.5 G filter. This system was calibrated with a reference solar cell (Model 15151, Abet Tech) to warrant a fixed light intensity of 200 mW cm$^{-2}$ in all of the experiments. Unless stated otherwise, the light was irradiated through the $BiVO_4$ semiconductor side (front-side). Photocurrents were measured while sweeping the potential from $-0.6$ to 0.7 V vs. Ag/AgCl in the positive direction with a scan rate of 10 mV s$^{-1}$. Electrochemical impedance spectroscopy (EIS) and Nyquist plot were performed using the same experimental setup as the PEC measurements, the frequency used was 40 mHz–100 kHz in 10 mV potential under continuous light.

### 2.7. Evaluation of Photocatalytic Activity

The photocatalytic activity of the semiconductor photocatalysts was initially evaluated for MB degradation. In each experiment, 25 mg of the photocatalyst was dispersed in 25 mL of aqueous methylene blue (10 mg.L$^{-1}$). Before the irradiation, the solution was magnetically stirred in dark for 30 min for MB adsorption on the surface of the photocatalyst. Subsequently, the solution was irradiated using a solar simulator equipped with a 150 W xenon lamp (Model 10500, Abet Tech) and an AM 1.5 G filter. This system was calibrated with a reference solar cell (Model 15151, Abet Tech) to warrant that light intensity was fixed at 300 mW cm$^{-2}$ in all of the photodegradation experiments. All of the experiments were conducted in triplicate, under continuous stirring, and aliquots of 300 μL were collected, centrifuged, and analyzed using a UV–VIS spectrophotometer. The concentration of the MB solution was monitored by the aliquot absorption at 664 nm.

Measurements of $O_2$ production were performed under visible light irradiation using 50 mg of photocatalyst suspended in 50 mL of 0.02 M $Fe(NO_3)_3.9H_2O$ as an electron scavenger. The air in the reactor vessel was evacuated to 60 Torr and purged with Argon repeatedly until all of the oxygen was eliminated from the reactor volume, which was confirmed by gas chromatography (SRI GC 8610C, SRI Instruments, Torrance, CA, USA). Then, the reactor was continuously irradiated with a 300 W Xe lamp located 18 cm from the

center of the reactor (light intensity of ~450 mW.cm$^{-2}$). A chemical filter of 0.22 M NaNO$_2$ was used to completely subtract the ultraviolet and some part of the infrared coming from the lamp.

## 3. Results and Discussion

The as-synthesized BVO particles were analyzed by FEG-SEM to investigate their morphological properties (Figure 1a). The images show irregular branched-shaped particles, with an estimated average size of 204 ± 66 nm, in agreement with previous studies [22,33]. The crystalline features of the pristine BVO sample were investigated by XRD analysis, as represented in Figure 1b. The X-ray diffraction data present narrow peaks that match that of the monoclinic scheelite phase (ICSD file 01751866), indicating the formation of well–crystalline monoclinic scheelite BVO, with a unit cell belonging to the spatial group I112/b, refined cell parameters of a = 5.18 Å, b = 5.08 Å, c = 11.67 Å, α = β = 90°, and γ = 90.36°, and a preferential orientation along with {040} facets, in accordance to the literature [33,36]. As expected, the XRD peaks of the Co-deposited BVO samples were identical to those of the pristine BVO due to the low Co content that is below the detection limit of the equipment (results not shown).

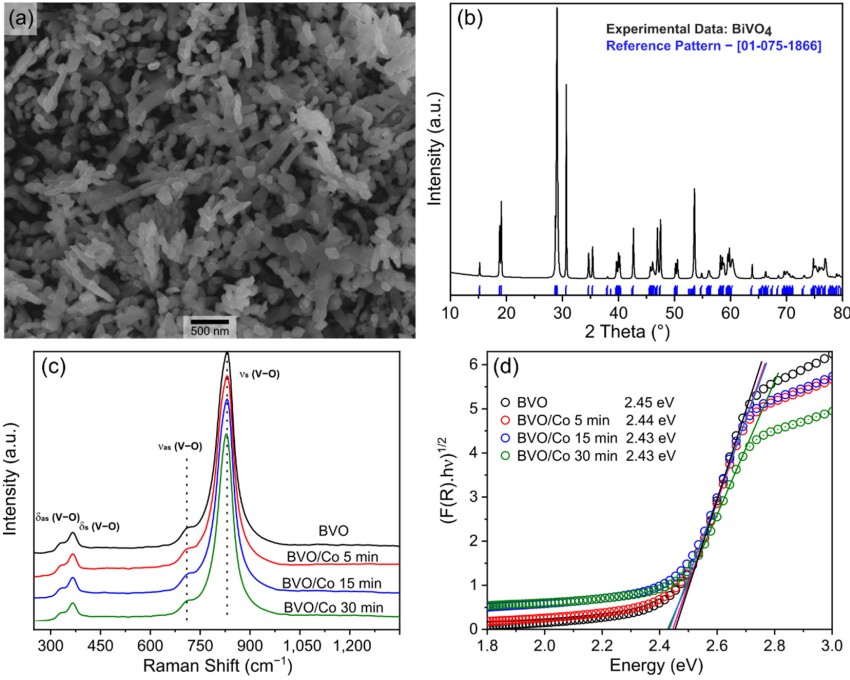

**Figure 1.** Characterization results of pristine BVO and BVO/Co powder samples: (**a**) FEG-SEM of pristine BVO; (**b**) XRD pattern of pristine BVO; (**c**) Raman spectra of pristine BVO and BVO/Co samples; (**d**) Tauc plot for the indirect bandgap determination of pristine BVO and BVO/Co.

Raman spectroscopy is an efficient tool for probing the structure and bonding in metal-oxide species through their vibrational characteristics. In the monoclinic phase of BiVO$_4$, the tetrahedral structure of VO$_4{}^{3-}$ is responsible for all BVO vibrational modes [37]. Figure 1c shows that the Raman spectra of the samples contain four main peaks, where the most intense one at 827 cm$^{-1}$ is attributed to the V–O symmetrical stretch mode. The other three peaks are located at 708, 366, and 325 cm$^{-1}$, and are attributed to the asymmetric stretching mode, and symmetrical and asymmetrical angular deformation mode, respectively, all of which are characteristics of the monoclinic phase as previously reported in the literature [37–40]. Additionally, it is possible to observe a slight shift in the main peak from 830 to 828 cm$^{-1}$ with increasing the Co deposition time (Figure 1c). However, according to the literature [22,40], we can estimate the V–O interatomic distance

for BVO, and it remained around 1.69 Å for all samples, showing that this small shift does not significantly impact the structural properties of the photocatalyst.

Optical properties were measured using UV–VIS diffuse reflectance spectroscopy (DRS). Figure 1d shows the obtained Tauc plots considering an indirect band gap. The estimated band gap for pristine BVO was 2.45 eV, which matched experimental values previously reported in the literature for monoclinic scheelite BVO [41–44]. All of the samples presented similar band gap energies, corroborating the fact that Co deposition did not change the structural properties of BVO, an outcome that shows the non-destructive character of the employed cocatalyst deposition route. However, one can see that the absorption of light increases in the visible range for higher Co sputtering times, as observed in the left tail of the curves, thus improving the light-harvesting properties of the photocatalysts.

The photocatalytic performance of the samples was initially investigated for MB degradation under visible light irradiation (Figure 2a). Before the photocatalytic reaction, the sample suspension was stirred for 30 min in the dark to reach the adsorption–desorption equilibrium of the MB. As can be seen, the efficiency of BVO/Co(5 min) photocatalyst overperformed all of the other studied samples, resulting in 78% of MB photodegradation after 60 min of simulated solar light irradiation, whereas using photocatalysts as pristine BVO, BVO/Co(15 min), and BVO/Co(30 min), the MB degradation was 70, 58, and 57%, respectively. For sputtering times higher than 5 min, the photocatalytic MB degradation decreased compared to pristine BVO, which may be due to the effects coming from the excess of Co, which can block light absorption through the main absorber material as well as reduce the number of surface active sites of BVO. Additionally, the photogenerate charge carriers in BVO might not be easily accessible for photocatalytic reactions in the presence of a thicker layer of Co between the surface of BVO and the solid/solution interface (see Table 1).

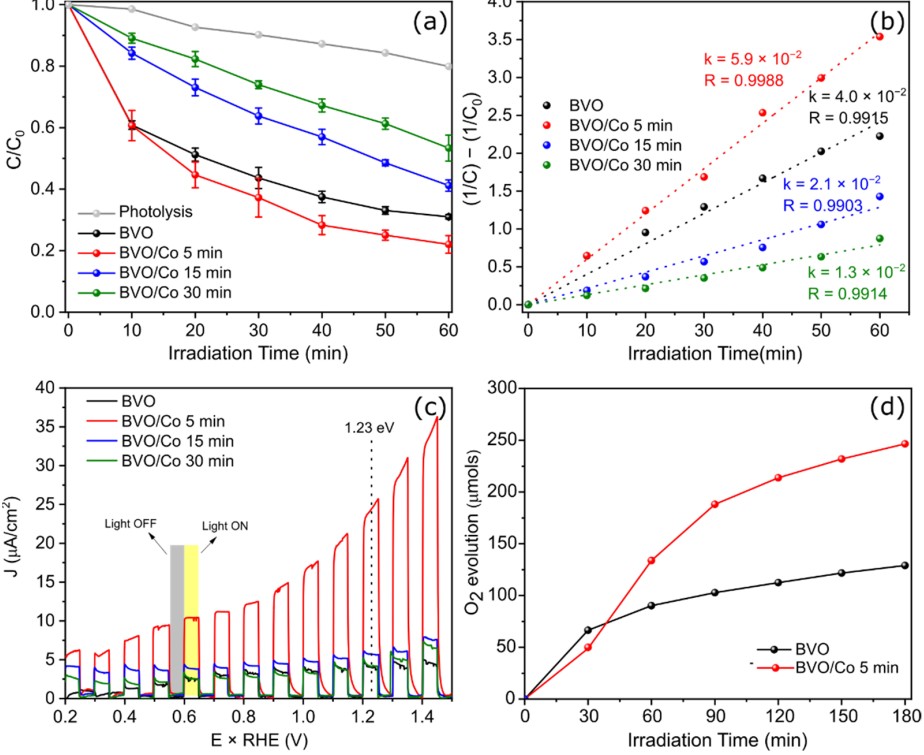

**Figure 2.** Photocatalytic performance of pristine BVO and BVO/Co samples under simulated solar irradiation (AM1.5G filter, 200 mW.cm$^{-2}$). (**a**) $C/C_0$ curves of the photocatalytic degradation of MB solution; (**b**) pseudo-second-order kinetic plots of MB solution degradation, where (k) is the pseudo-second-order kinetic constant and R is the correlation coefficient of the linear fit; (**c**) PEC measurements using a 0.1 M of $Na_2SO_4$ electrolyte; (**d**) $O_2$ production via water oxidation.

**Table 1.** Values of $R_{ct}$, $R_s$, and CPE obtained from the EIS spectra of BVO and BVO/Co($x$ min) samples.

| SAMPLES | BVO | BVO/Co(5 min) | BVO/Co(15 min) | BVO/Co(30 min) |
|---|---|---|---|---|
| $R_{ct}$ (K$\Omega$) | 91.9 | 17.3 | 76.4 | 101.8 |
| $R_s$ ($\Omega$) | 22 | 22 | 22 | 22 |
| CPE-P | 0.90 | 0.85 | 0.91 | 0.92 |
| CPE-T ($\mu$F) | 25.3 | 32.7 | 17.2 | 14.1 |

Figure 2b shows the value of the pseudo-second-order kinetic constants (k) for MB removal in the presence of BVO and different BVO/Co photocatalysts. The kinetic constant of pristine BVO is 0.040 min$^{-1}$ against 0.059 min$^{-1}$ for BVO/Co(5 min), showing that the deposited Co accelerated the MB removal by a factor of ~1.5 times. Figure 2c shows the chopped LSV of the BVO/Co photoanodes under PEC water oxidation conditions. All photoelectrodes were light sensitive, as expected, but only a small photocurrent response was observed for the pristine BVO under light conditions, returning to near zero current in the dark (Figure 2c). The BVO/Co(5 min) photoanode presented the highest photocurrent response of 23 $\mu$A cm$^{-2}$ at 1.23 V vs. RHE, compared with only 5 $\mu$A.cm$^{-2}$ for pristine BVO. Larger Co sputtering times resulted in lower photocurrents. Even though the photocurrent is not competitive with the best results reported for BVO in the literature [45,46], this result shows that an improvement of almost 4.6 times in the photocurrent can be obtained by using naked Co-based NP cocatalyst deposition. This improvement is aligned with other reports of BVO-based photoanodes, where modification of the surface with an oxygen evolution cocatalyst (OER) is central for accelerating the water oxidation kinetics and decreasing surface charge recombination [34]. This result indicates that the density of photoholes transferred from the semiconductor to the liquid medium to promote water oxidation is higher for the sample of BVO/Co(5 min), as the charge carrier recombination at the semiconductor surface was probably reduced [35].

The water oxidation performance of the powder photocatalysts for oxygen evolution was investigated using iron nitrate as the electron acceptors, Figure 2d. The results show that the pristine BVO presented an $O_2$ production of 112 $\mu$mols after 180 min of simulated solar irradiation. The BVO/Co(5 min) photocatalyst, however, presented a much higher $O_2$ evolution in the same period, reaching a more than two-fold increase in the total amount of evolved $O_2$. These results indicate that the deposition of the naked Co-based cocatalyst onto the BVO surface is a promising strategy for enhancing the photocatalytic activity for the photooxidation of water.

EIS measurements with and without light irradiation were performed on the photoelectrodes to investigate the charge transfer resistance from the photoanode to the electrolyte. The EIS results are presented through the Nyquist plot in Figure 3a which presented single semicircle characteristics and could be simulated using the equivalent Randles circuit shown in the inset of Figure 3a, which includes a series solution resistance ($R_s$) from the electrolyte, a resistance of charge transfer at the interface between the working electrode and the electrolyte ($R_{ct}$), and a constant phase element (CPE) [47–49]. The last is necessary to describe the double layer capacitance distribution across the inhomogeneous electrode surface. The final best-fitted results can be seen in Table 1. As a result, the deposition of Co for 5 min significantly improved the charge transfer at the semiconductor/electrolyte interface by reducing $R_{ct}$ by a factor of ~4.3 compared with the pristine BVO, increasing the PEC water oxidation efficiency (Figure 2c).

Photoluminescence (PL) is a useful technique for studying electron–hole recombination ($e^-$–$h^+$) in semiconductor materials, as the PL signal is the result of electron decay from an excited state to some lower energy state or ground state [50]. The intensity of the PL light emitted is directly related to the type/structure of the materials [51] and the recombination rate of $e^-$–$h^+$, where a lower emitted PL light (at a fixed wavelength near the edge of the absorption band) can be indicative of minor $e^-$–$h^+$ recombination [52,53]. Thus, when looking at Figure 3b, the PL light emitted by BVO/Co(5 min) is smaller compared with the pristine BVO sample, thus indicating lesser $e^-$–$h^+$ recombination.

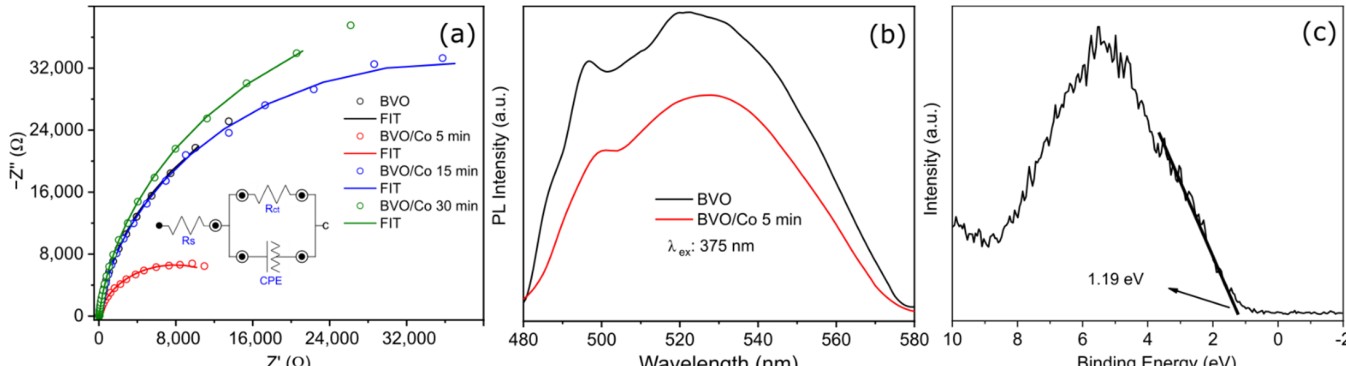

**Figure 3.** Electrochemical, photoluminescence, and valence band characterization; (**a**) Nyquist plot of the pristine BVO and BVO/Co samples; (**b**) PL emission spectra under 375 nm excitation; (**c**) XPS valence band spectra of pristine BVO.

XPS measurements were performed to investigate the surface chemical properties of the BVO/Co catalysts, the valence band maximum energy (VB) with respect to the Fermi level ($E_F$), and to estimate the amount of Co deposited onto BVO. From the valence band high-resolution spectra, the VB was determined by extrapolating the edge of the XPS peak to its intersection with the background [54]. The VB of pure BVO was determined at 1.19 eV below the $E_F$ (Figure 3c), which is in good agreement with the results from the literature [55].

The Bi 4f and V 2p XPS spectra of the pristine BVO sample were previously reported by our group and showed only lines for $Bi^{3+}$ and $V^{5+}$ oxidation states, as expected [22,33]. It is important to mention that for cobalt speciation and quantification, the measurements were firstly performed in the BVO/Co(5 min) sample, but the low amount of cobalt resulted in spectra of a poor quality, even for higher counting times. Therefore, we prepared a BVO/Co(45 min) sample especially for conducting Co speciation through high-resolution XPS measurement. Figure 4 shows the O 1s and Co 2p3/2 high-resolution XPS spectra for the pristine BVO and BVO/Co(45 min).

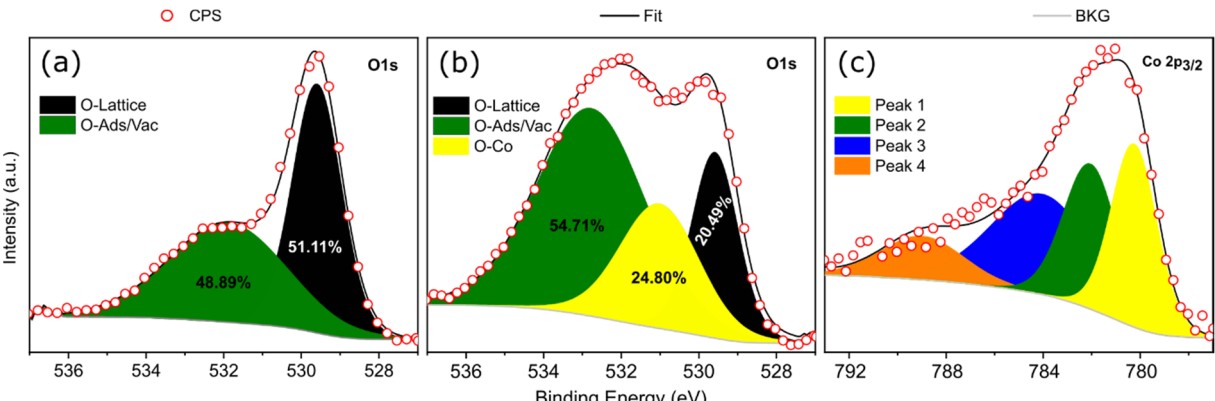

**Figure 4.** High-resolution XPS spectra of (**a**) O 1s of pristine BVO sample; (**b**) O 1s; (**c**) Co 2p3/2 of BVO/Co(45 min) sample.

The pristine BVO showed two peaks in the O 1s region (Figure 4a), located at 529.6 (51.11%) and 532.2 eV (48.89%), attributed to the oxygen species of the BVO crystal lattice and the presence of oxygen vacancies, adsorbed C-O-H species, or hydroxyl groups on the surface, respectively [56–58]. Conversely, the BVO/Co presented three O 1s peaks located at 529.6 (20.49%), 531.0 (24.80%), and 532.8 (54.71%) eV (Figure 4b). The first and the last had identical attributions to those previously identified for pristine BVO and the additional peak located at 531.01 eV, in turn, could be attributed to the O-Co bond [59,60] and confirmed the presence of oxidized cobalt at the photocatalyst surface. However, it

an area contribution near 1:1 for the peaks at 529.6 and 532.8 eV would be expected, as obtained for pristine BVO. Therefore, some additional contribution is convoluted at the 532.8 eV peak after Co deposition. As the sputtering process is not expected to increase the surface oxygen vacancies in BVO, we may assume that the contribution of adsorbed hydroxyl groups (or C-O-H species) was higher in the BVO/Co photocatalyst surface.

Bazylewski et al. described that samples containing Co tend to oxidize when exposed to moisture, especially for ultrasmall-sized particles, which can form CoO or $Co(OH)_2$ [61]. To further validate our hypothesis, we analyzed the Co 2p3/2 high-resolution XPS spectrum of BVO/Co (Figure 4c). It shows an intense peak (Peak 1) centered at 780.2 eV and three additional satellite peaks (noted as 2, 3, and 4) located at higher binding energies. These sets of peaks were attributed to $Co^{2+}$ species. It is worth noting that $Co^{2+/3+}$ ($Co_3O_4$) and $Co^0$ main peaks would be expected at 779 and 778 eV, respectively [60,62,63], but were not identified. Therefore, combining the O 1s and Co 2p findings, we can infer that the Co oxidized to $Co(OH)_2$ [61,64] after the sputtering process and the unavoidable exposure to ambient moisture. Table 2 shows a summary of all of the fitted peak positions and their FWHM.

**Table 2.** The XPS peak position of the samples and % weight and atomic of Co onto BVO.

| Samples | O 1s | | | | Co 2p3/2 | | |
|---|---|---|---|---|---|---|---|
| | Peak | Binding Energy (eV) | FWHM (eV) | Area (%) | Peak | Binding Energy (eV) | FWHM (eV) |
| **BVO** | O_Lattice | 529.60 | 1.48 | 51.11 | - | - | - |
| | O_Ads/Vac | 531.90 | 3.55 | 48.89 | - | - | - |
| **BVO/Co (45 min)** | O_Lattice | 529.58 | 1.41 | 20.49 | Peak 1 | 780.23 | 2.2 |
| | O-Co | 531.02 | 2.44 | 24.80 | Peak 2 | 782.10 | 2.6 |
| | O_Ads/Vac | 532.78 | 3.31 | 54.71 | Peak 3 | 785.89 | 5.0 |
| | - | - | - | - | Peak 4 | 790.57 | 4.0 |

Assuming that the sputtering deposition rate is constant over time, a linear behavior of the amount of sputtered Co with the deposition time was expected, as previously observed by quantitative analysis [35]. We then estimated the surface Co coverage on BVO by analyzing the survey spectrum of the BVO/Co(45 min) sample, obtaining 2.98 wt.% of Co at the surface, and by linear regression, where 0.33, 0.99, and 1.99 wt.% were projected for the 5, 15, and 30 min samples. Although it is not the overall composition, photocatalysis is a surface-related phenomenon and thus the most important topic of discussion is the surface coverage of the photocatalyst.

To further rationalize our results, we built up the energy diagram of the BVO/Co photocatalyst system. As the work function of the BVO was reported at about 5.6 eV [65] and the XPS valence band spectrum revealed that the VB was located at 1.19 eV below $E_F$, the VB was located at 6.79 eV in the vacuum scale. This value was 2.35 V on the normal hydrogen electrode (NHE) scale ($E_{NHE} = E_{abs} - 4.44$, at 298 K). Therefore, the CB was then located at $-0.1$ V vs. NHE as the obtained bandgap energy was 2.45 eV, and the energy diagram and the possible charge transfer mechanisms for the $BVO/Co(OH)_2$ system is proposed in Figure 5. The CB and VB values of $Co(OH)_2$ respectively assumed at $-1.54$ and 1.31 V, respectively, vs. NHE, based on reports from the literature [66,67].

In all of the cases, both semiconductor materials could be active under visible light irradiation and electrons could be excited from their VB to their CB, leaving $h^+$ in the VB. Of course, the photoexcitation process was dominant for BVO particles as the best photocatalyst had only 0.33 wt.% of $Co(OH)_2$ as the surface content. If a type-II heterojunction was formed (Figure 5a), photogenerated $h^+$ were transferred from the VB of the BVO to the VB of $Co(OH)_2$, where they may partially oxidize $Co^{2+}$ to $Co^{3+}$. Subsequently, $Co^{3+}$ could oxidize water (or MB pollutant) and become reduced back to its former oxidation state ($Co^{2+}$), re-establishing the cycle so that it could be oxidized again by the next photogenerated $h^+$ [66,68,69]. The alternation between the $Co^{2+}$ and $Co^{3+}$ species would help

in the charge separation process, improving the overall photoactivity. However, in such a path, the electrons would accumulate at the BVO surface and recombine back with $h^+$ at the BVO VB, as observed for the pristine BVO, which should diminish the photoactivity of the BVO/Co system and not the opposite. It is important to note that in the case of photocatalytic oxygen evolution using iron nitrate as the electrons acceptors, the type-II heterojunction, to some extent, explains the observed results, as the accumulated electrons at BVO may be consumed to reduce $Fe^{3+}$ to $Fe^{2+}$, avoiding $e^- - h^+$ recombination at the BVO photocatalyst. The same could be rationalized for PEC water oxidation, where electrons from BVO were conducted to the FTO substrate and, in sequence, to the counter electrode across the external circuit. For the MB photodegradation, however, the type-II heterojunction mechanism failed. Particularly for this last photocatalysis, electrons trapped at the BVO surface would not have enough driving force to generate $O_2^{\bullet-}$ radicals ($E^0$ $(O_2/O_2^{\bullet-})$ = $-0.33$ V) [22], playing against increasing the photoactivity.

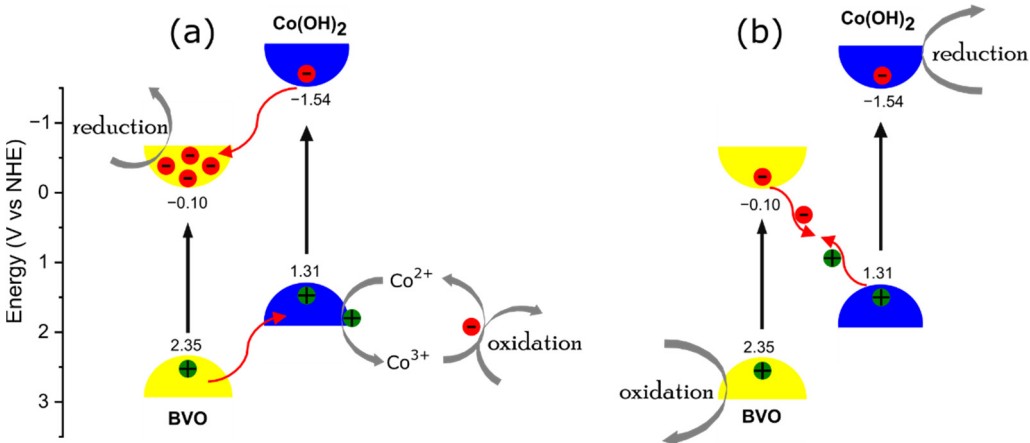

**Figure 5.** Proposed mechanisms of charge transfer at the BVO/Co(OH)$_2$ photocatalysts: (**a**) type-II and (**b**) direct Z-scheme (S-scheme) heterojunction.

Therefore, another probable mechanism of charge transfer is a direct Z-scheme (or S-scheme), as depicted in Figure 5b, where electrons previously excited to the CB of BVO directly recombine with holes at the VB of Co(OH)$_2$, keeping holes and electrons physically separated and at their highest redox potentials [2,70]. In such a way, photogenerated holes at the BVO/liquid interface would conduct the oxidation reactions directly (water and MB) or by hydroxyl radical formation ($E^0$ ($OH^-/^\bullet OH$) = +1.9 V vs. NHE) [22], while electrons at the CB of Co(OH)$_2$ would easily perform the reduction reactions (water, $O_2/O_2^{\bullet-}$, iron nitrate, and MB). Therefore, the S-scheme is the mechanism that better explains all of the suspended powder photocatalytic processes described in this work, i.e., MB photodegradation and oxygen evolution using iron nitrate electron acceptors.

The presence of the Co-based cocatalysts, therefore, improves visible light absorption by the dual absorption process and, at the same time, decreases the $e^- - h^+$ recombination and improves the charge transfer at the semiconductor/electrolyte interface.

## 4. Conclusions

A special vibrating apparatus was successfully employed to sputter coat different amounts (0.33–1.99 wt.%) of Co-based NPs cocatalysts on the surface of scheelite monoclinic BVO powders using different deposition times (5–30 min). As observed, the sputtering deposition did not significantly alter the BVO structure, and the XPS analysis indicated the formation of Co(OH)$_2$ on the surface of the BVO particles, showing that the as-deposited metallic Co oxidized upon exposure to ambient air. Compared with pristine BVO, the optimized BVO/Co(5 min) photocatalyst showed a (i) higher photoactivity in the photocatalytic degradation of MB; (ii) an almost 400% increase in photocurrent for PEC water oxidation; and (iii) a more than two-time higher O$_2$ evolution through water oxidation,

all under simulated solar irradiation. The deposition of $Co(OH)_2$ cocatalysts not only increased visible absorption, but also reduced the electron–hole recombination, as well as facilitated charge transfer at the solid/solution interface. The BVO/Co system thus performed better than pristine BVO in terms of oxygen evolution, PEC water oxidation, and pollutant photodegradation reactions. The magnetron sputtering-based deposition strategy reported here is general and can be extended to other photocatalysts/cocatalyst systems and guide the development of more efficient photocatalytic systems for diverse photo(electro)chemical applications.

**Author Contributions:** The manuscript was written through the contributions of all of the authors. L.E.G.: methodology, investigation, writing—original draft. L.F.P.: investigation, writing—original draft. W.S.R.: investigation. R.V.G.: investigation, writing—review and editing, formal analysis. S.U.: investigation, writing—review and editing, formal analysis. H.W.: conceptualization, supervision, project administration, investigation, writing—review and editing, funding acquisition. All authors have read and agreed to the published version of the manuscript.

**Funding:** This study was financed in part by the Fundação Universidade Federal de Mato Grosso do Sul, UFMS/MEC, Brazil; by the Coordenação de Aperfeiçoamento de Pessoal de Nível Superior, Brasil (CAPES), Finance Code 001; by the National Council for Scientific and Technological Development (CNPq); and by The Foundation for Support to the Development of Teaching, Science, and Technology of the State of Mato Grosso do Sul (FUNDECT).

**Institutional Review Board Statement:** Not applicable.

**Informed Consent Statement:** Not applicable.

**Data Availability Statement:** Not applicable.

**Acknowledgments:** The authors would greatly thank CAPES (L.E.G. funding), CNPq (Projects 486342/2013-1, 427835/2016-0, 311798/2014-4, 313300/2020-8, and 310066/2017-4), and FUNDECT (N. 099/2014, 106/2016, and 228/2022). W.S.R. thanks the CNPq fellowship # 152738/2022-3. L.E.G. thanks the CNPq fellowship # 300715/2021-8, 301035/2021-0, and 302245/2021-9. S.U. acknowledges higher education commission (HEC) Pakistan for financial assistance (Project # 9286). The authors also acknowledge Marco A.U. Martines for the X-ray diffraction measurements.

**Conflicts of Interest:** The authors declare no conflict of interest. The funders had no role in the design of the study; in the collection, analyses, or interpretation of data; in the writing of the manuscript; or in the decision to publish the results.

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
