# Peer review of "Increasing the Photocatalytic Activity of BiVO4 by Naked Co(OH)2 Nanoparticle Cocatalysts"

_2673-7256, doi:10.3390/photochem2040055_

Round 1

Reviewer 1 Report

This paper describes the production procedure and photocatalytic performance of hybrid catalysts based on bismuth vanadate coated with cobalt hydroxide nanoparticles. The resulting catalysts demonstrate a significantly increased photoactivity both in dye photodegradation and water photooxidation compared to the pure BVO photocatalyst. This effect is shown to be due to the increased visible light absorption of the modified catalysts, promoted charge transfer at the interface, and reduced electron/hole recombination. Heterogeneous photocatalysts obtained are characterized by SEM, XRD, XPS, UV-Vis DRS, Raman and photoluminescence spectroscopy. The conclusions made are rather reliable and fully supported by the experimental data. The results obtained by different methods are discussed in details, and a possible mechanism of the effects observed is proposed. Hence, the paper is worth publishing in Photochem, and there are just minor corrections to be made before the paper can be accepted for publication:

- Line 293: Fig. 3d should be Fig. 2d

- Line 313: replace a comment with a reference.

Author Response

We are very thankful to the reviewer for his/her time to read our manuscript, valuing the importance of the topic. We made sure to carefully revise the text, further improving the discussion on the subject and clarifying some minor details. We hope the reviewer will be more satisfied with the revised version.

-Line 293. We double-checked Figure 3d citation and it is already correct. To avoid misinterpretation and following the Reviewer's suggestion we have included a mention of “powder photocatalyst” to distinguish it from Figure 2c (photoelectrochemical water oxidation).

- Line 313. An error occurred when converting MS Word to PDF in the system so the link to Table 1 was broken. We thank the Reviewer for the observation. It is now corrected in the revised version.

Reviewer 2 Report

This manuscript investigated the Cocatalytic effect of Co for BVO in visible light photodegradation of MB. It is worth to be published after some revision.

1. the XRD characterization is incomplete, they only gives one sample of BVO4;

2. SEM is not enough to distinguish the "Co nanoparticles", TEM shall be added.

3. The source of Co NPs does not be mentioned.

 4. There are some grammar mistakes

Author Response

Please, find our responses in the attached file.

Reviewer 3 Report

The present manuscript deals with the study of photocatalytic activity of BiVO4 by naked 2 Co(OH)2 nanoparticle cocatalysts. The manuscript is interesting, all the experiments were well performed and presented. In my opinion, it should be revised.

Comments,

1) Novelty of the work is not clear.

2) In the introduction section, the following reports should be discussed and cited, ref. FlatChem 24, 100200, 2020; International Journal of Hydrogen Energy 46 (68), 33696-33717, 2021, 2021; Journal of Environmental Management 268, 110677, 2020.

3) 2.2 Synthesis of BVO nanoflakes can be given as flowchart.

4) n Error! Not a valid 313 bookmark self-reference?? Included in the manuscript, should be rectified.

5) 3. Results and Discussion, subsections should be included.

6) Conclusions should be rewritten. 

Author Response

(The authors gave the same response as above.)

Round 2

Reviewer 2 Report

The authors have answered the questions properly. It can be accepted as it is.